# The Relationship between Fluoride Exposure and Cognitive Outcomes from Gestation to Adulthood—A Systematic Review

**DOI:** 10.3390/ijerph20010022

**Published:** 2022-12-20

**Authors:** Banu Preethi Gopu, Liane B. Azevedo, Ralph M. Duckworth, Murali K. P. Subramanian, Sherley John, Fatemeh Vida Zohoori

**Affiliations:** 1School of Health &and Life Sciences, Teesside University, Middlesbrough TS1 3BX, UK; 2School of Human and Health Sciences, University of Huddersfield, Huddersfield HD1 3DH, UK

**Keywords:** fluoride, children, cognitive outcome, systematic review

## Abstract

Chronic exposure to high levels of fluoride may cause health concerns, including in cognitive function. This study reviewed the evidence on the association between fluoride exposure and cognitive outcomes in children from gestation up to 18 years old. A literature search was conducted for studies on pregnant women and children below 18, exposed to any source of fluoride, and assessed with a validated cognitive tool. The data were analyzed using a systematic narrative synthesis approach and by subgroup: study design, age of participants, levels of fluoride exposure and methodological quality. Our search retrieved 15,072 articles, of which 46 met the inclusion criteria. Only 6 of the studies had a longitudinal design; the remainder were cross-sectional. The levels of fluoride exposure were ≥2 mg/L in 27 studies and <2 mg/L in 13 studies; 6 studies did not report levels of fluoride exposure. Only 1 of 5 studies graded as excellent quality showed a negative association between fluoride exposure and cognitive outcomes, whereas 30 of 34 poor and fair quality studies reported a negative association. The overall evidence from this review suggests that high fluoride exposure might be associated with negative cognitive outcomes in children. However, more longitudinal studies with high methodological quality are needed on this topic.

## 1. Introduction

Fluoride is the ionic form of fluorine, a trace element with a unique ability to inhibit and even reverse dental caries by promoting enamel remineralization and constraining acid production by plaque bacteria [1]. Community-based fluoride delivery strategies, such as water-, salt- and milk-fluoridation schemes, have been adopted over many decades for preventing dental caries, especially in areas where the fluoride levels in water are low [2]. Community water fluoridation (CWF) is the most cost-effective method for the prevention of dental caries, which has been implemented in 25 countries worldwide [3,4]. 

Although naturally or artificially fluoridated water at optimal levels (0.7–1.0 mg F/L) improves dental health, exposure to high levels of fluoride could result in dental or skeletal fluorosis. The environmental protection agency (EPA) of the US National Research Council set the maximum acceptable concentrations of fluoride in drinking water to 2 mg/L to prevent dental fluorosis and 4 mg/L to prevent skeletal fluorosis [5,6]. 

While fluorosis is a well-recognized adverse effect of excessive fluoride exposure, the scientific basis for its adverse non-dental health outcomes is contradictory and inconclusive. In particular, there is evidence from systematic reviews and meta-analyses conducted among humans, animals, and cell culture lines, both linking and refuting the role of fluoride in neurodevelopmental disorders in children [7,8,9,10,11]. Brain development is influenced by genetic expression and natural factors, and the disruption of either could fundamentally alter neural function [12]. It is known that brain development begins in the third gestational week [12], and its size increases four-fold during the preschool period, reaching approximately 90% of the adult volume by the age of six [13]. The developing brain is much more susceptible to injury caused by toxicants than the mature brain, which may lead to permanent damage [14]. However, a comprehensive synthesis of the existing literature is needed in particular to explore the effect of fluoride levels on different population age groups.

Our review is preceded by two recent reviews on the same topic. The national toxicology program (NTP), which is part of the US Department of Health and Human Services report [11], is a large and comprehensive review which included studies conducted in humans, animals, and in vitro. That report, however, did not explore the effect of fluoride exposure according to age group, exposure level, and study design. Likewise, a review by Miranda et al. [10] had broader inclusion criteria, including children and adults, while the current review focuses only on studies conducted among pregnant women and children up to 18 years old. Equally, the Miranda et al. review [10] included all neurological disorders but only reported IQ associations, limiting the generalisability of the effects of fluoride on different cognitive domains. In our review, we include all studies which used a validated cognitive tool. Furthermore, Miranda et al. [10] only included cross-sectional studies, while studies adopting a longitudinal study design should be included to provide knowledge on the dynamic process of change over time on children’s development. Finally, an update from previous reviews is needed, as more studies around the topic of interest have appeared since the other reviews were published. 

Therefore, the overall aim of this systematic review is to synthesize the evidence on the effect of fluoride exposure during pregnancy and through to young adulthood (up to 18 years of age) on cognitive outcomes. This systematic review is novel as it updates previous systematic reviews, includes longitudinal studies, and examines the differences according to population, fluoride exposure, study design, and study quality.

## 2. Materials and Methods

This systematic review was guided by the Cochrane handbook for systematic reviews [15] and the Joanna Briggs Institute (JBI) manual for evidence synthesis [16] and is reported using the updated preferred reporting items for systematic reviews and meta-analyses (PRISMA) criteria [17]. The protocol for this systematic review was registered in the international prospective register of systematic reviews, PROSPERO (reg. no. CRD42021230649).

### 2.1. Selection Criteria

This review included:Research studies conducted among pregnant women and children below the age of eighteen;Studies with participants directly exposed to fluoride through sources including groundwater, tea and milk, or indirectly exposed through breakfast cereals, seafood, toothpaste, mouthwash, industrial emissions, coal-burning for fuel, supplements, pesticide residues, and certain pharmaceuticals;Studies focusing on the association between fluoride exposure and cognitive outcomes in children below eighteen years;Only studies with a validated tool to assess the cognitive outcomes;Longitudinal, cross-sectional, and experimental studies;Only publications in the English language.

Studies conducted on animals and human participants above eighteen years of age were not included. Also, case studies, narrative reports, expert opinions, reviews, abstracts without full texts and conference presentations were not eligible for inclusion.

### 2.2. Search Strategy

This review adopted a peer-reviewed dual-step search strategy. Initially, a scoping search was conducted in MEDLINE and PubMed with key terms such as: “Fluoride”, “Children”, “Mother”, “Cognitive outcomes”, “IQ”, “ADHD”, “Focussed attention”, “Sustained attention”, and “Academic achievements” to scope the available literature. Several articles from this initial search were explored to expand the search terms and develop a more rigorous search strategy. The search terms and the strategy were peer-reviewed by the research librarian of Teesside University and the authors, and the full search was performed in June 2021 using seven electronic databases: MEDLINE, Embase, and CINHAL via EBSCO host, PubMed, Web of Science, Scopus, and PsycINFO using MeSH (medical subject headings) terms (Appendix A). We also screened the references cited in the included studies to identify studies that may not have been picked up in the electronic search.

The references were uploaded to Endnote (Version X9.0), and digital and manual deduplication was performed. The first author screened the titles and abstracts of all the articles identified from the search, and two authors (MKPS and SJ) each screened 50% of the articles. The title and abstract screening resulted in 69 articles being screened at the full-text level. The first author screened all the articles, and the rest of the team screened 20% each. Discrepancies were solved among all authors through discussions.

### 2.3. Data Extraction

The data from the included studies were entered into a customized data extraction form. A pilot data extraction was performed by BPG, FVZ, LBA and RMD with two articles to assess the feasibility of the data extraction sheet. The entire data extraction was completed by the primary author and cross-checked by the other authors (FVZ, LBA and RMD). The following data were extracted: study ID, country, study design, setting, total subjects, ethnicity, socioeconomic status, subgroup size, mean age of children, percentage of males, pregnancy/postpartum status, source of fluoride exposure, duration of fluoride exposure, fluoride category, fluoride concentration, confounding factors, cognitive assessment tool, validation reference, cognitive outcomes, correlation variable assessed, correlation, adjustment covariates, mean differences, statistical analysis, and statistical inference.

### 2.4. Quality Assessment

The methodological quality of the included studies was assessed using the modified version of the Strengthening the Reporting of Observational Studies in Epidemiology (STROBE-M) cross-sectional and cohort rating tool [18]. The first author assessed the quality of all the included studies, and the other authors (FVZ, LBA and RMD) assessed 33.3% each. The STROBE-M, a modified version, provides general reporting recommendations for descriptive observational studies and studies investigating associations between exposures and health outcomes. The 22-item scores range from 1 to 4, with the total possible score being 77 for cross-sectional studies. We used Limaye et al. [18] publication quality grades for the STROBE-M checklist to classify the studies into four categories: ≥85%: excellent, 70–85%: good, 50–70%: fair, and <50%: poor. For the purpose of this review, studies of any quality were included.

### 2.5. Data Synthesis and Analyses

The data were synthesized using narrative synthesis [19]. The main characteristics of the eligible studies, such as the participant characteristics, source of fluoride exposure and level of exposure, outcome measures, and mean IQ score, were combined in a summary table (Appendix A) and accompanied by an overview of the systematic review characteristics and findings. A narrative synthesis was performed initially, providing a descriptive synthesis of all the included studies. A subgroup analysis was then performed to explore the association between fluoride exposure and cognitive outcomes, offering an overview of the significance of the results and the direction of the effect. The following subgroups were included:Study design: We synthesized the data on two study designs, i.e., longitudinal and cross-sectional;Participant age group: The studies were grouped according to age range and divided into study populations of ≤8 and >8 years old. These thresholds were selected with reference to the Centre for Disease Control (CDC), which classifies the first eight years as the period when learning, health, and success are mostly developed [20];Fluoride level: The studies were grouped into concentrations < 2 mg/L and ≥2 mg/L. This threshold was based on the U.S. EPA recommended maximum level of 2 mg/L of fluoride in drinking water to prevent enamel fluorosis [21];Study quality: The studies were assessed for methodological quality using the classification of Limaye et al. [18].

## 3. Results

A total of 31,335 articles were identified from the database search. After deduplication, 15,072 records were screened at the title and abstract level, and a total of 69 articles were included in the full-text screening based on the pre-set inclusion and exclusion criteria. From those, 46 studies were eligible and included in the review. The study selection process is presented in Figure 1, depicting the PRISMA flow diagram.

A summary of the extracted data, including information on author, year, country, study design, sample size, age, gender, level of fluoride exposure, and type of cognitive outcome measurement, is provided in Appendix A.

### 3.1. Study Countries, Population Age Group, and Study Design

Of the 46 studies included in this review, 50% of the articles (*n* = 23) were conducted in China [22,23,24,25,26,27,28,29,30,31,32,33,34,35,36,37,38,39,40,41,42,43,44]. The next highest contribution was from India, with nine articles [45,46,47,48,49,50,51,52,53]. Six studies were published from Mexico [54,55,56,57,58,59], three articles were from Canada [60,61,62], and two were from Iran [63,64]. One article each was published fromMongolia [65], New Zealand [66] and Pakistan [67].

A total of 21,501 participants were included across the 46 studies, with 52.8% of them being male. The age of children included in 44 studies ranged from 3 to 17 years, while two of the studies [28,56] were conducted among infants with ages ranging from 28 to 41 weeks. Among the included studies, only six had a longitudinal study design [24,54,55,56,61,66]. The remaining studies (*n* = 40) used a cross-sectional design.

### 3.2. Fluoride Exposure Route, Level, and Duration

From the included studies, the two sources of fluoride exposure of participants were drinking water and coal burning. Only four studies [27,29,36,38] included participants exposed to fluoride through coal burning, while participants in the other studies were exposed to fluoride through their drinking water. The fluoride levels to which the participants were exposed ranged from 0.13 to 9.4 mg/L in the drinking water. The levels of fluoride exposure through coal burning ranged from 0.03 to 2.33 mg/m^3^. The majority of the studies (*n* = 28) reported exposure to fluoride from birth. In contrast, a number of studies (*n* = 18) did not report the duration of exposure [27,28,29,32,34,36,37,38,39,41,48,52,55,57,60,64,65,67]. In the majority of the studies (*n* = 41), the authors assessed the correlation between the different levels of fluoride exposure and the cognitive development of the participants as the primary outcome. In the other five studies [32,37,41,58,67], the correlation was performed as a secondary outcome analysis.

### 3.3. Type of Cognitive Outcome Measurement

Among the 46 included studies, 24 used an original or adapted version of Raven’s standard progressive matrices, and 11 studies [23,29,32,34,55,57,58,61,62,66,67] used the Wechsler intelligence scale. Other tools used to assess the cognitive status of the participants were: official intelligence quotient (IQ) tests [44], the Chinese comparative scale of intelligence test [41], the Raymond B Cattell test [63], the Chinese Binet IQ test [27], wide range assessment of memory and learning [23], wide range assessment of visual motor ability [23], Conners’ continuous performance test [33,54], McCarthy scales of children’s abilities, and the Canadian health measures survey questionnaire [55]. Two of the included studies [28,56] reported neonatal cognitive assessments using specialized tools, namely the Bayley scale of infant development II (BSDI-II) and the standard neonatal behavioural neurological assessment (NBNA).

The cognitive assessment tools define cognitive status using different units. The Raven’s standard progressive matrices, the Wechsler intelligence scale, the Raymond B Cattell test [63], the Chinese Bidet IQ test [27], and the Conners rating scale [33,54] provide mean IQ levels. A few other studies reported cognitive status using different units rather than IQ, including the strengths and difficulties questionnaire [24], the developmental coordination disorder (DCD) scale [24], the mental development index, and the psychomotor development index [56]. A few studies [31,36,50,52,59] reported the distribution of intelligence rankings using a modified version of Raven’s standard progressive matrices.

### 3.4. Methodological Quality of the Included Studies

Out of the 46 studies included in the review, five [25,40,55,60,61] scored more than 85%, indicating excellent quality. Seven studies [35,43,54,56,59,62,66] were of good quality, scoring between 70 and 85%, and 14 studies [23,24,26,29,33,37,39,42,45,47,48,49,51,67] scored between 50 and 70%, rendering them fair. The remaining 20 studies scored less than 50% and were, therefore, of poor quality. The detailed scores for each individual study are presented in Appendix A.

### 3.5. Outcome Analysis

In the outcome analysis, we synthesized the data on the association between total fluoride exposure on cognitive outcomes from all studies and conducted a subgroup analysis.

#### 3.5.1. Overall Analysis

A total of 31 out of the 46 included studies reported their cognitive outcomes using mean IQ scores alone. Of these, 25 [22,27,29,30,32,33,34,35,37,39,42,43,44,45,47,48,49,51,53,58,61,62,63,64,67] concluded that the mean IQ levels of children exposed to fluoride at more than or equal to 2 mg/L were significantly lower than those exposed to <2 mg/L, while the remaining 6 studies reported no significant association between the fluoride exposure and the mean IQ of participants [40,41,46,54,65,66].

Ten studies reported outcomes such as the mental and psychomotor development index [56], neonatal behavioural neurological assessment scores [28], intelligence ranking [31,36], mean intelligence grades [50,59], and intelligence assessment scores [25,26,52,57]. These studies showed a significant negative association between fluoride exposure and the measured cognitive outcome. Studies that reported outcomes such as self-reporting learning ability [60], the mean general cognitive index [55], the strengths and difficulty questionnaire [24], or intelligence deficiency [38] showed no effect, whereas one study which reported the outcome through a wide-ranging assessment of memory and learning [23], showed a significant negative effect.

#### 3.5.2. Subgroup Level Analysis

We report results using a subgroup analysis according to (1) study design; (2) age group; (3) fluoride level; and (4) study quality (Table 1).

##### Study Design

From the six longitudinal studies included in this systematic review, three studies (50%) [54,56,61] identified a significant negative association between fluoride exposure and cognition. While from the 40 cross-sectional included studies, 34 studies (85%) showed a significant negative association between fluoride and cognition.

##### Age Group

Eleven studies [23,24,28,34,54,55,56,58,60,61,62] were conducted with participants up to eight years old, one study did not report participant age [57], and the remaining 34 studies included participants of ages more than eight years. Among the studies conducted in children aged eight and younger, eight studies (72%) [23,28,34,54,56,58,61,62] reported a significant negative association between fluoride and cognition, and among the 9–18 year group, 28 studies (82%) reported a significant negative association.

##### Fluoride Level

Of the 11 studies that reported fluoride exposure levels below 2 mg/L [26,35,40,52,54,55,59,60,61,62,66], six studies (54%) [26,35,52,54,61,62] reported a significant negative association between fluoride exposure and children’s cognition, two of which were longitudinal studies [54,61]. The majority of the studies (*n*= 26) were conducted among fluoride exposure levels equal to or above 2 mg/L, of which 24 showed a significant negative association between fluoride and cognition. However, only one of these studies [56] had a longitudinal design. Nine of the studies included in this review [24,29,32,33,36,38,48,57,65] did not provide the levels of fluoride exposure of their participants.

##### Study Quality

From the five studies classified as of excellent quality [25,40,55,60,61], only one study (20%) [61] reported a significant negative association between fluoride exposure and cognitive outcome in children, and this study had a longitudinal design. Out of the seven studies [35,43,54,56,59,62,66] which were classified as good quality, two longitudinal studies [54,56] and three cross-sectional studies [35,43,62] showed a significant negative relationship between fluoride and cognition. Among the 14 studies [23,24,26,29,33,37,39,42,45,47,48,49,51,67] that scored “fair”, 13 studies [23,26,29,33,37,39,42,45,47,48,49,51,67] had a significant negative association, and from the 20 studies [22,27,28,30,31,32,34,36,38,41,44,46,50,52,53,57,58,63,64,65] that were classified as poor quality, 17 studies [22,27,28,30,31,32,34,36,44,50,52,53,57,58,63,64,65] reported a significant negative association.

## 4. Discussion

This systematic review synthesized the evidence on the association between fluoride exposure and cognitive outcomes in children from gestation up to 18 years of age. Out of the 46 included studies, only 5 were considered of excellent methodological quality, of which four reported no association between fluoride and cognition, whereas a higher percentage of the poor quality studies showed a negative association.

Furthermore, the majority (87%, *n* = 40) of the included studies in our systematic review were cross-sectional studies. A cross-sectional study captures a population at a single point in time and hence is not capable of establishing cause and effect. Therefore, a cross-sectional study is not an ideal tool to assess the impact of chronic exposure to fluoride on changes in developmental outcomes, such as cognitive development.

Our subgroup analysis, based on age group (≤8 and 9–18 years), showed that the impact of fluoride exposure on cognition appeared to be similar for each age group. This is mainly due to the belief that the critical window for cognitive development is the first three years of life, as profound changes in brain development are known to occur during this period [68].

Our systematic review showed that the negative association between fluoride exposure and cognitive outcomes appears to be stronger at high levels of fluoride exposure (≥2 mg/l) compared to lower levels (<2 mg/l): 92% of studies at higher levels compared to 54% of studies at lower levels. Our findings are in agreement with the conclusions of the two relevant systematic reviews on this topic. The NTP review [11], which included studies conducted among humans, animals, and in vitro, concluded that the effects on cognitive neurodevelopment were inconsistent at concentrations of 0.03- 1.5 mg/L; and the Miranda et al. review [10] cited a lack of evidence to support that fluoride exposure is associated with any neurological disorder.

A major limitation of human studies on the health impact of fluoride is the lack of well-documented fluoride exposure in the studied populations. In almost all the included studies in our review, the fluoride concentrations of drinking water were used as a proxy of fluoride exposure, and the fluoride intake from other sources was not considered. The main sources of fluoride exposure in children are diet and unintentional ingestion of fluoridated toothpaste. The contribution of water (as a drink) to the total daily fluoride intake could be as low as 4% in children younger than six years of age [69,70], whereas fluoridated toothpaste could account for up to 87% of the total daily fluoride intake [71].

An important point to note is that child cognitive development is complex and could be influenced by several physiological and environmental factors. It was estimated that, globally, 200 million children younger than five years old fail to attain their development potential, mainly due to poverty, nutritional deficiencies, and inadequate learning opportunities [72]. Major risk factors for poor cognitive development include: intrauterine growth restriction, stunting, deficiencies of iodine and iron, malaria, exposure to lead, HIV, maternal depression, and inadequate cognitive stimulation. In contrast, protective factors include maternal education and breastfeeding [72,73].

Although 61% (*n* = 28) of the included studies in our systematic review considered protective factors, only one study [58] included a major risk factor (lead). None of the included studies considered other important influencing factors in cognitive development, such as iodine deficiency, which represents the greatest single cause of brain damage globally. It was associated with a global loss of 10–15 IQ points at a population level [74]. According to the World Health Organisation [75], nearly two billion people, including 285 million school-age children, are iodine deficient across the globe. It has been suggested that deficiencies of other nutrients, such as selenium, iron and vitamin A, might also intensify the effects of iodine deficiency [74]. In particular, endemic dietary selenium deficiency has been reported in some parts of the world, most particularly in China [76,77].

Half of the included studies in our systematic review were from China. This large number could reflect the concern in that country regarding the safety and impact of groundwater fluoride on health, as levels of groundwater fluoride reach up to 15 mg/L in some parts of China [78]. It is important to highlight that iodine deficiency was recorded as a significant public health problem in the Chinese population in 1995, with over 700 million people being iodine deficient [79]. According to a study conducted in Chongqing in 1994, a high percentage (41.5%) of schoolchildren aged 7–14 years were iodine deficient [79]. The Chinese government started a policy of salt iodization in 1995 which resulted in the “almost” elimination of iodine deficiency by 2000. Therefore, studies conducted on the effect of fluoride on cognitive development in China before 2000 should be interpreted with caution.

Finally, the descriptive data of this review showed that, out of the 46 included studies that assessed cognition in children, 31 had reported this as mean IQ levels. This is potentially important, as cognition is a complex area of study and limiting it by only assessing IQ scores is a shortcoming [80].

### 4.1. Strengths and Limitations

This review was conducted and presented following the PRISMA guidelines to report the findings from a systematic review, and the review protocol was registered and prepared as publicly available on PROSPERO, rendering the process transparent and reliable. The search strategy was meticulously devised with several iterations in consultation with an academic librarian and the entire research team. The search was conducted using several scientific databases with very few limiters, such as language, ensuring it covered a vast field and that no eligible study was missed. Another strength of the review is the inclusion criterion permitting studies only if they used a validated tool to assess the cognitive outcome, increasing the trustworthiness of the findings.

One limitation of the review is the inclusion of all eligible studies despite their scoring in the quality assessment, and the overall conclusions of this review might be adversely influenced due to many of the included studies being classified as fair or poor quality. Another limitation is the lack of a meta-analysis due to the inconsistent reporting of the cognitive outcomes across different studies using various assessment tools and units.

### 4.2. Recommendations for Future Studies

Considering the sensitive nature of this research area and consequent ethical concerns, experimental studies with human participants are highly unlikely. However, this review has highlighted that the majority of the studies (87%) have a cross-sectional design and 73% are of fair or poor quality, limiting the interpretation. Careful mechanistic studies and robust epidemiological studies must be conducted in order to provide further insights into the possible association between fluoride exposure and cognition. In order to better assess causality within the observed relationship, future observational studies should have a longitudinal design and consider developing models to include all factors which could contribute to cognitive development in children.

## 5. Conclusions

The overall evidence from this systematic review suggests that exposure to fluoride at a level of more than 2 mg/L in drinking water may result in impaired cognitive outcomes among children. However, the inclusion of many low quality studies and the lack of robust estimates of fluoride exposure from all sources make it difficult to provide definitive conclusions. It is essential to select the appropriate tool to assess the different domains of cognition, and future studies must take a more robust approach, use longitudinal designs and also explore the role of fluoride in the broader parameters of cognition.

## Figures and Tables

**Figure 1 ijerph-20-00022-f001:**
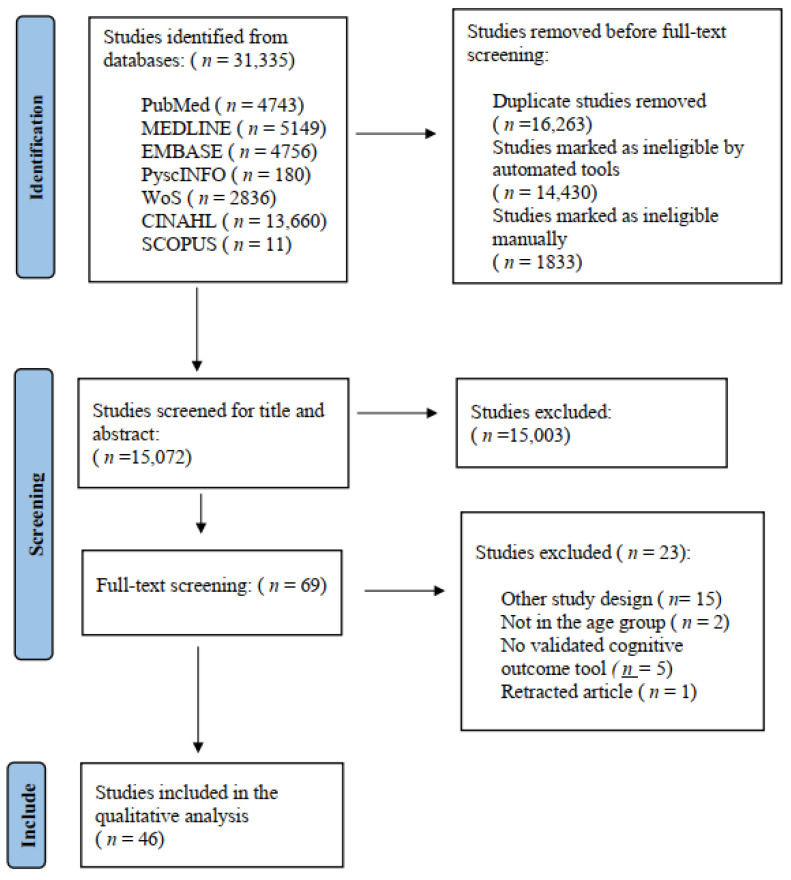
PRISMA flow diagram depicting the process of article screening.

**Table 1 ijerph-20-00022-t001:** Number of studies by subgroup analysis.

Subgroups	All Studies		Studies Reporting a Significant Negative Relationship between Fluoride Exposure and Cognition.	
	*n*	Study Reference	*n* (%)	Study Reference
**Study design**				
Longitudinal	6	[24,54,55,56,61,66]	3 (50)	[54,56,61]
Cross-sectional	40	[22,23,25,26,27,28,29,30,31,32,33,34,35,36,37,38,39,40,41,42,43,44,45,46,47,48,49,50,51,52,53,57,58,59,60,62,63,64,65,66,67]	34 (85)	[22,23,25,26,27,28,29,30,31,32,33,34,35,36,37,39,42,43,44,45,47,48,49,50,51,52,53,57,58,62,63,64,65,67]
**Age group**				
≤8 yr	11	[23,24,28,34,54,55,56,58,60,61,62]	8 (72)	[23,28,34,54,56,58,61,62]
>8 yr	34	[22,25,26,27,29,30,31,32,33,35,36,37,38,39,40,41,42,43,44,45,46,47,48,49,50,51,52,53,59,63,64,65,66,67]	28 (82)	[22,25,26,27,29,30,31,32,33,35,36,37,39,42,43,44,45,47,48,49,50,51,52,53,63,64,65,67]
Not reported	1	[57]	1 (100)	[57]
**Fluoride level**				
<2 mg/L	11	[26,35,40,52,54,55,59,60,61,62,66]	6 (54)	[26,35,52,54,61,62]
≥2 mg/L	26	[22,23,25,27,28,30,31,34,37,39,41,42,43,44,45,46,47,49,50,51,53,56,58,63,64,67]	24 (92)	[22,23,25,27,28,30,31,34,37,39,42,43,44,45,47,49,50,51,53,56,58,63,64,67]
Not reported	9	[24,29,32,33,36,38,48,57,65]	7 (77)	[29,32,33,36,48,57,65]
**Study quality**				
Excellent	5	[25,40,55,60,61]	1 (20)	[61]
Good	7	[35,43,54,56,59,62,66]	5 (71)	[35,43,54,56,62]
Fair	14	[23,24,26,29,33,37,39,42,45,47,48,49,51,67]	13 (93)	[23,26,29,33,37,39,42,45,47,48,49,51,67]
Poor	20	[22,27,28,30,31,32,34,36,38,41,44,46,50,52,53,57,58,63,64,65]	17 (85)	[22,27,28,30,31,32,34,36,44,50,52,53,57,58,63,64,65]

## Data Availability

All data generated or analyzed during this study are included in this article and its supplementary material files. Further enquiries can be directed to the corresponding author.

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
