# Peer review of "The Relationship between Fluoride Exposure and Cognitive Outcomes from Gestation to Adulthood—A Systematic Review"

_ijerph, 2022, doi:10.3390/ijerph20010022_

Round 1

Reviewer 1 Report

Dear Authors

Thank you for this wonderful study.

Four concerns

1) Are the conclusions consistent with the evidence and arguments presented, hence GRADE analysis is suggested?

2) Please include the quality assessment figure or table in the manuscript.

3) Recommendation / suggestion of a model to assess all factors that assess child's cognitive growth can be made.

4) GRADE analysis can be done

Author Response

Thank you for this wonderful study.

Four concerns

Comment (1) Are the conclusions consistent with the evidence and arguments presented, hence GRADE analysis is suggested?

Response (1)

Our conclusion is based on our findings and evidence from our findings. However, we believe that  GRADE analysis is not appropriate for this review since this is not a review of interventions and we were not able to perform a meta-analysis. Please refer to Response 4 below for detailed explanations.

Comment (2) Please include the quality assessment figure or table in the manuscript.

Response (2)

Thank you for this comment. Due to its size, we have not included the quality assessment table within the body of the manuscript. However, we have now included the table as a supplementary file (Supplementary file 3) and cited it in the text (Page 6, lines 220-221 and page 9, lines 379-380).

Comment (3) Recommendation / suggestion of a model to assess all factors that assess child's cognitive growth can be made.

Response (3)

We have now added this recommendation to Section 4.2 (Page 9, lines 365-368).

Comment (4) GRADE analysis can be done

Response (4)

As explained in “Section 4.1”, we could not perform a meta-analysis due to the heterogeneity of the data collected. From our knowledge, GRADE analysis is performed in reviews that have conducted a meta-analysis, and the effect size can be inputted (Malmivaara, A., 2015. Methodological considerations of the GRADE method. Annals of Medicine, 47(1), pp.1-5.).

Although we were not able to rate the certainty of evidence using GRADE, we assessed the quality of each included study by using the modified Strengthening the Reporting of Observational Studies in Epidemiology (STROBE - M) quality assessment tool, which has been extensively used.

Reviewer 2 Report

The present manuscript is an exhaustive systematic review of the relationship between fluoride exposure and cognitive outcomes. The manuscript is well written and structured, making it easier to understand. The systematic review evaluates the quality of the included studies, ensuring the overall quality of the systematic review. However, in the results section, the manuscript only describes the results of the included literature based on subgroups and no further analysis was performed.

1. The manuscript divides the included studies into two groups based on fluoride exposure levels, but some studies assessed fluoride in drinking water, while others examined fluoride in urine. It is not rigorous to compare fluoride from different sources.

2. This systematic review might have made the conclusions more adequate if meta-analysis had been available.

Author Response

Comment (1) The present manuscript is an exhaustive systematic review of the relationship between fluoride exposure and cognitive outcomes. The manuscript is well written and structured, making it easier to understand. The systematic review evaluates the quality of the included studies, ensuring the overall quality of the systematic review. However, in the results section, the manuscript only describes the results of the included literature based on subgroups and no further analysis was performed.

Response (1)

Thank you for the positive comments. Concerning the information in the results session, we provided a descriptive overview of all studies at the start (items 3.1, 3.2, 3.3, 3.4). We then performed an outcome analysis, including overall and subgroup analysis. We have now also added the additional analysis suggested by you (Comment 2) concerning the data synthesis of fluoride from different sources. 

Comment (2) The manuscript divides the included studies into two groups based on fluoride exposure levels, but some studies assessed fluoride in drinking water, while others examined fluoride in urine. It is not rigorous to compare fluoride from different sources.

Response (2)

Thank you for this comment. To address this comment, we have now revised Table 1/fluoride level subgroup (page 6) and relevant text (results – page 7, lines 259-266; and discussion – page 8, lines 296-297): i.e. we have now separated studies in which fluoride in drinking water was reported with those that did not report it.

We have also amended column headings and contents in Supplementary File 2 [Methods of assessment of Fluoride exposure, and Water fluoride level at baseline (mg/L)] for better clarity.

Comment (3) This systematic review might have made the conclusions more adequate if meta-analysis had been available.

Response (3)

We agree with the reviewer. However, as explained in “Section 4.1”, due to the heterogeneity of the data collected, we were not able to perform a meta-analysis.

Nevertheless, we performed a narratively synthesised subgroup analysis and used a robust quality assessment tool (i.e. STROBE - M) to synthesis and report the findings of the included studies.

Reviewer 3 Report

1.     Considering that an appropriate dose of fluoride may be beneficial to the body, I am curious whether any studies included in this review have explored the nonlinear relationship between fluoride exposure and intelligence?

2.     Whether the author would like to discuss the current social concern about the neurological effects of fluoride. As far as I know, in some countries with drinking water fluoridation, some official background organizations have been resisting the study on the negative correlation between fluoride and intelligence.

Author Response

Comment (1) Considering that an appropriate dose of fluoride may be beneficial to the body, I am curious whether any studies included in this review have explored the nonlinear relationship between fluoride exposure and intelligence?

Response (1)

Since none of the included 46 studies (both cross-sectional and longitudinal) reported a non-linear relationship between the fluoride levels and cognitive outcome, one can assume either “there was no non-linear relationship” or “the authors of those paper did not explore the non-linear relationship”.

Comment (2) Whether the author would like to discuss the current social concern about the neurological effects of fluoride. As far as I know, in some countries with drinking water fluoridation, some official background organizations have been resisting the study on the negative correlation between fluoride and intelligence.

Response (2)

Thank you for this comment. We are aware of these issues. However, discussing the social concern about the neurological effects of fluoride will alter the scope of this review. Hence, we decided not to mention those.

Round 2

Reviewer 2 Report

No more comments.